

# Validation of an algorithm to assess regular and irregular gait using inertial sensors in healthy and stroke individuals

Carmen Ensink[1,2], Katrijn Smulders[1], Jolien Warnar[1] and Noel Keijsers[1,2,3]

[1] Department of Research, Sint Maartenskliniek, Nijmegen, the Netherlands
[2] Department of Sensorimotor Neuroscience, Donders institute for Brain, Cognition and Behaviour, Radboud University, Nijmegen, the Netherlands
[3] Department of Rehabilitation, Donders institute for Brain, Cognition and Behaviour, Radboud Univeristy Medical Center, Nijmegen, the Netherlands

Corresponding authors
Carmen Ensink,
c.ensink@maartenskliniek.nl
Katrijn Smulders,
k.smulders@maartenskliniek.nl

## ABSTRACT

**Background.** Studies using inertial measurement units (IMUs) for gait assessment have shown promising results regarding accuracy of gait event detection and spatiotemporal parameters. However, performance of such algorithms is challenged in irregular walking patterns, such as in individuals with gait deficits. Based on the literature, we developed an algorithm to detect initial contact (IC) and terminal contact (TC) and calculate spatiotemporal gait parameters. We evaluated the validity of this algorithm for regular and irregular gait patterns against a 3D optical motion capture system (OMCS).

**Methods.** Twenty healthy participants (aged $59 \pm 12$ years) and 10 people in the chronic phase after stroke (aged $61 \pm 11$ years) were equipped with 4 IMUs: on both feet, sternum and lower back (MTw Awinda, Xsens) and 26 reflective makers. Participants walked on an instrumented treadmill for 2 minutes (i) with their preferred stride lengths and (ii) once with irregular stride lengths ($\pm 20\%$ deviation) induced by light projected stepping stones. Accuracy of the algorithm was evaluated on stride-by-stride agreement of IC, TC, stride time, length and velocity with OMCS. Bland-Altman-like plots were made for the spatiotemporal parameters, while differences in detection of IC and TC time instances were shown in histogram plots. Performance of the algorithm was compared between regular and irregular gait with a linear mixed model. This was done by comparing the performance in healthy participants in the regular vs irregular walking condition, and by comparing the agreement in healthy participants with stroke participants in the regular walking condition.

**Results.** For each condition at least 1,500 strides were included for analysis. Compared to OMCS, IMU-based IC detection in both groups and condition was on average 9–17 (SD ranging from 7 to 35) ms, while IMU-based TC was on average 15–24 (SD ranging from 12 to 35) ms earlier. When comparing regular and irregular gait in healthy participants, the difference between methods was 2.5 ms higher for IC, 3.4 ms lower for TC, 0.3 cm lower for stride length, and 0.4 cm/s higher for stride velocity in the irregular walking condition. No difference was found on stride time. When comparing the differences between methods between healthy and stroke participants, the difference between methods was 7.6 ms lower for IC, 3.8 cm lower for stride length, and 3.4 cm/s lower for stride velocity in stroke participants. No differences were found on differences between methods on TC detection and stride time between stroke and healthy participants.

**Conclusions**. Small irrelevant differences were found on gait event detection and spatiotemporal parameters due to irregular walking by imposing irregular stride lengths or pathological (stroke) gait. Furthermore, IMUs seem equally good compared to OMCS to assess gait variability based on stride time, but less accurate based on stride length.

## INTRODUCTION

Gait analysis is a valuable tool for diagnosis and treatment evaluation of gait impairments in clinical settings. Traditionally, optical motion capture systems (OMCS) and force plates are used for gait analysis. However, a major downside of these expensive systems is the requirement of a lab with specialized staff, limiting the accessibility of gait analysis in clinical practice. Moreover, these dedicated labs often provide optimal controlled conditions for gait assessment, limiting ecological validity (*Renggli et al., 2020*). A promising alternative for OMCS-based gait analysis is the use of inertial measurement units (IMUs). These compact wearable sensors enable gait analysis not restricted to the lab setting, at a lower cost, and easier to operate.

In the past decades, many research groups have developed algorithms for gait assessment based on IMU data. An overview of different algorithms is given in a reviewing article by *Díaz, Stephenson & Labrador (2019)* and a performance comparison of seventeen common algorithms is made in the review by *Pacini Panebianco et al. (2018)*. These gait algorithms are developed to identify gait events and subsequently calculate spatiotemporal gait parameters. Despite the increasing number of studies in this field, several limitations hinder further uptake in clinical settings.

The first obstacle to using IMU-based gait parameters in the clinic stems from a scarcity of validation studies testing IMU-based algorithms in people with irregular walking patterns. The validity of gait algorithms has predominantly been tested for regular gait in healthy participants (*Pacini Panebianco et al., 2018*). However, individuals with gait deficits often walk with irregular patterns (*e.g.*, due to neurological diseases) (*Pacini Panebianco et al., 2018*; *Mariani et al., 2013*; *Sabatini et al., 2005*). It is known that data from IMUs is less predictable in irregular gait patterns, compromising the performance of many of the developed algorithms (*Hundza et al., 2014*, *Yang et al., 2013*). For example, *Hundza et al. (2014)* found a mean error of two cm in stride length estimation in healthy controls, which increased to 11 cm in people with Parkinson's disease.

A second limitation of published algorithms is that they are typically designed and optimized for specific locations of the IMUs on the body, *e.g.*, shank, ankle or shoe. As signal features of IMUs depend on the body location, the applicability of these algorithms to placement on other body parts can be limited. Most common set-ups are (combinations of) one IMU on the lower back, one sensor on each shank, or on both feet (*Pacini Panebianco et al., 2018*). In general, IMUs placed closer to the source of impact (lower

legs or feet with the walking surface) have the most prominent signal features (*Pacini Panebianco et al., 2018*; *Washabaugh et al., 2017*). Moreover, *Jasiewicz et al. (2006)* found that feet-based algorithms outperform shank-based algorithms regarding the accuracy of gait event detection in pathological gait. They concluded that the irregular and less smooth movement of the shank during pathological gait was likely due to increased instability, which in turn caused more disturbances in the sensor signal (*Jasiewicz et al., 2006*). Unfortunately, the number of studies evaluating the validity of gait algorithms processing data from IMUs on the feet in pathological gait is limited (*Díaz, Stephenson & Labrador, 2019*; *Pacini Panebianco et al., 2018*).

Finally, almost all validated gait algorithms are undisclosed. So far, most gait algorithms are only schematically described in published articles (*Mariani et al., 2013*; *Sabatini et al., 2005*; *Hundza et al., 2014*; *Yang et al., 2013*; *Jasiewicz et al., 2006*; *Behboodi et al., 2019*; *Carcreff et al., 2018*; *Fadillioglu et al., 2020*; *Mariani et al., 2010*; *Trojaniello et al., 2014*; *Teufl et al., 2019*). Without code sharing, replication, validation, and use of these algorithms in the clinic remains challenging.

Based on previous literature (*Sabatini et al., 2005*; *Behboodi et al., 2019*; *El-Gohary et al., 2014*; *Mercer et al., 2003*), we developed an algorithm for gait assessment using IMUs on both feet and the trunk (lumbar level). We evaluated the validity of this algorithm against an OMCS for regular and irregular walking patterns. We operationalized irregular walking patterns in two ways: first, by using stepping targets on an instrumented treadmill in healthy participants, cueing walking with constant and varying step lengths. Secondly, we evaluated the algorithms in people with stroke. Based on results previously reported in the literature we based our algorithm on (*Sabatini et al., 2005*; *Behboodi et al., 2019*; *El-Gohary et al., 2014*; *Mercer et al., 2003*), we expected a similar and small constant error of less than five cm between the gait algorithm and OMCS in regular and irregular walking in healthy participants and in regular walking in people with stroke. However, a larger variability of the error in the stroke population compared to the healthy population, was expected based on previous literature in pathological gait (*Caldas et al., 2017*). Participants were tested on a treadmill to collect a large number of steps for each participant. Healthy participants performed overground walking to ensure consistent results for this condition. We also developed the gait algorithm in an open-source programming language (Python), making the data and code freely available for further use.

## MATERIALS & METHODS

### Participants

Healthy participants aged between 40 and 90 years old, who were able to walk for at least two minutes without assistance were recruited from the community between April 2021 and February 2022. We included five participants per age category, 40–49, 50–59, 60–69 and 70+ years, resulting in a total of $N = 20$ healthy participants (Table 1). Exclusion criteria were any diseases affecting gait or balance, such as osteoarthritis, neurological or neuromuscular disease or deformities of the lower extremities, and BMI >30 kg/m$^2$.

Stroke participants were able to walk for at least two minutes without assistance, participated in a walking therapy group due to their stroke, were above 18 years, and had to

**Table 1** **Participant characteristics.** FAC: Functional Ambulation Categories.

|  | Healthy participants | Stroke population |
|---|---|---|
| N | 20 | 10 |
| Gender (male/female) | 10/10 | 7/3 |
| Age (mean ± SD years) | 59 ± 12 | 61 ± 11 |
| Height (mean ± SD cm) | 174 ± 7.2 | 176 ± 7.5 |
| Weight (mean ± SD kg) | 75 ± 8.0 | 81 ± 9.1 |
| Affected side (left/right) | – | 4/6 |
| Stroke type (ischemic/hemorrhagic) | – | 7/3 |
| FAC score (min–max) | – | 3–5 |

understand verbal instructions. Exclusion criteria were any other diseases affecting gait or balance, hemispatial neglect, and a BMI >30 kg/m$^2$. A total of $N = 10$ stroke patients from the gait rehabilitation program of the Sint Maartenskliniek were included in this study. Clinical data of the stroke participants were derived from the electronic patient record. Participant characteristics can be found in Table 1, the Chi-square and Mann–Whitney U tests revealed no significant differences ($p \geq 0.05$) between the groups based on gender, age, height or weight.

The study protocol was in line with the Declaration of Helsinki and was granted an exemption of the Dutch medical scientific research act (WMO) by 'METC Oost-Nederland' (identification number: 2021-8191). Prior to study participation, written informed consent was provided by all participants.

## Materials

Participants were equipped with 4 IMUs (MTw Awinda, Xsens, Enschede): at the dorsal side of both feet, sternum, and lower back (L4/5) and 26 reflective markers for the OMCS according to the VICON plug-and-gait lower body model (*Vicon, 2021*). Xsens MT Manager software suite version 2019.2 was used for data capture of the IMUs. All treadmill walking conditions were performed on the GRAIL (Gait Real-time Interactive Analysis Lab, (Motek Forcelink, Amsterdam, the Netherlands)); an instrumented treadmill with an ten-camera OMCS (VICON, Oxford, United Kingdom). All overground walking conditions were performed in the overground gait lab, with a ten-camera OMCS (VICON, Oxford, United Kingdom). The IMU system and OMCS recorded at 100 Hz and were synchronized by a high-low pulse with OMCS as master.

## Measurement protocol

During treadmill walking, healthy and stroke participants wore a harness attached to the ceiling for safety precautions. All participants walked on the treadmill at a self-selected speed before data collection for approximately four minutes for familiarization purposes. Subsequently, they walked on the instrumented treadmill in two conditions: regular and irregular treadmill walking.

In the regular treadmill walking condition, participants walked for 2 min at a self-paced, comfortable walking speed. Self-paced walking allowed participants to adjust the speed of

the treadmill by walking more at the front of the belt (increase in speed) or at the back of the belt (decrease in speed). The participants with stroke performed the regular walking task in self-paced mode if possible, but in fixed speed mode if their walking capacity was insufficient to regulate the treadmill's speed. After the regular walking condition, all participants performed the irregular walking condition, consisting of a precision stepping task at the average walking speed during the regular walking condition. Rectangular stepping stones (30x15 cm) were projected on the treadmill, functioning as step targets. The stepping stones were projected at stride lengths randomly varying between 80–120% (80, 90, 100, 110, 120) of the preferred stride length of the individual participant. Participants were instructed to walk without holding the handrail bars if possible but were allowed to do so if needed. Participants were allowed to rest between walking conditions. For each of the measurements, data collection was started once participants indicated they had reached a comfortable walking speed. Data were recorded for a duration of two minutes and stopped before participants were decelerating, ensuring no accelerating and decelerating phases were included in the dataset for further analysis.

The healthy participants performed an additional overground walking task. They were asked to walk ten times back and forth through the measurement volume of the overground gait lab (approximately 5 m) at a comfortable walking speed.

## Data processing

All data processing and analysis described in this paragraph and in 'Data analysis' are included in the algorithm code available at: https://github.com/SintMaartenskliniek/IMU_GaitAnalysis (Release 'Validation study', tag 'v1.1.0').

IMU data captured by MT Manager software (2019.2) included angular velocity and acceleration in the sensor frame, acceleration in the earth frame, and orientation in quaternion and Euler angle format. OMCS data was captured by VICON Nexus software (version 2.4). All further data processing and analyses were performed in Python 3.10.

Prior to any data analysis, a second-order low-pass Butterworth filter was applied to the angular velocity (15 Hz cut-off frequency) and acceleration data (17 Hz cut-off frequency) of the IMUs, according to *Sabatini et al. (2005)*. OMCS data was filtered with a second-order low-pass Butterworth filter (15 Hz cut-off frequency).

## Data analysis

Figure 1 shows a typical gait cycle and corresponding mediolateral gyroscope and vertical accelerometer signals of the IMUs on the feet, and the velocity of the heel and toe markers of the OMCS. A gait cycle consists of a stance phase, initiated by an initial contact event (IC), and a swing phase, starting at a terminal contact event (TC). Accurate identification of IC and TC events is crucial for correctly calculating spatiotemporal gait parameters. They also define the time period for further integration of the IMU signals to determine spatial parameters.

Another important gait event for the IMU-based gait algorithm is the instant of mid-swing, which is used to identify the IC and TC. Based on *Sabatini et al. (2005)*, mid-swing events were identified as the maximum in the clockwise direction of the angular velocity
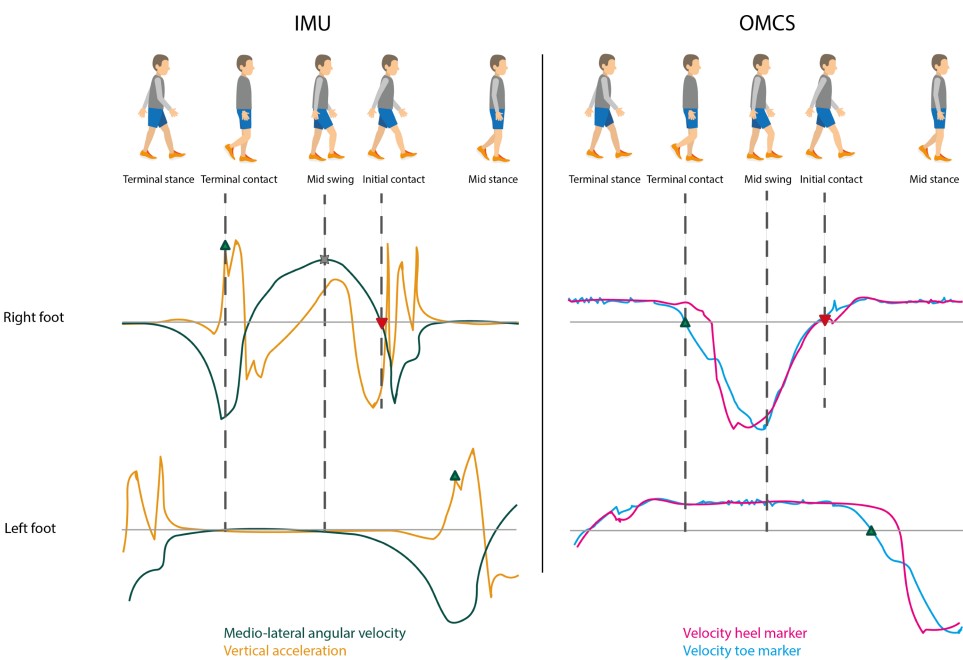

**Figure 1** **Typical gait cycle with corresponding IMU and OMCS data.** Upper graph is of the right foot and the lower graph is of the left foot as presented in the gait cycle on the top of the figure. The left graphs show the angular velocity around medio-lateral axis (flexion-extension movement, green) and vertical acceleration (earth frame, orange) of the IMUs attached to the feet. The right graphs show the velocity of the toe marker (blue) and velocity of the heel marker (pink) of the OMCS. Terminal contact (green triangle pointing up) was determined at the peak acceleration before mid-swing (IMU) and zero-crossing of the velocity of the toe marker (OMCS). Mid-swing (cross) was identified at peak angular velocity (IMU). Initial contact (red triangle pointing down) was identified at the zero-crossing of the angular velocity after mid-swing (IMU) and zero-crossing of the velocity of the heel marker (OMCS). Foot flat was identified between terminal contact and mid-swing of the contralateral foot (for both IMU and OMCS).

around the mediolateral axis (*i.e.,* the axis of rotation for flexion-extension movements). To this end, scipy.signal.find_peaks function with peak distance at 0.7 s, prominence at 1 rad/s was used. Based on *Behboodi et al. (2019)*, IC events were identified at the first instance of zero-crossing, positive to negative, after mid-swing in the angular velocity around the mediolateral axis. Peaks between mid-swing events in the vertical acceleration based on *Mercer et al. (2003)* were used to identify TC events (scipy.signal.find_peaks function with no further specifications) (*Mercer et al., 2003*; *Mo & Chow, 2018*). In case multiple peaks were found, the peak at the instance with the largest angular velocity in the anti-clockwise direction was identified as the actual TC event; the others were deemed as an artefact. Finally, foot flat was identified based on *Behboodi et al. (2019)*. The start of foot flat was defined as the instant of TC on the contralateral side. The end of foot flat (*i.e.,* heel-off) was identified at the instant of mid-swing on the contralateral side (*Behboodi et al., 2019*). Based on the instants of the gait cycle, stance phase (initial contact to terminal contact), swing phase (terminal contact to initial contact) and foot flat phase (start of foot flat to end of foot flat) were identified.

After gait event detection, spatial parameters were calculated. The tri-axial velocity of the foot was estimated by numerical integration of the accelerometer (earth frame) signal according to Eq. (1) over the duration of the trial (120 s). However, this involves some signal drift. To reduce this signal drift, a sigmoid curve, based on a p-chip interpolation (scipy.interpolate.pchip_interpolate function) was subtracted from the signal (zero-velocity updates) (*Mariani et al., 2010*). The p-chip interpolation function was defined between each period of foot flat (Eq. (2)). Hereafter, zero-velocity updates were applied at each foot flat (Eq. (3)) (*Sukumar, 2010*; *Mohamed Refai et al., 2020*; *Wu et al., 2021*).

$$\text{velocity}_{\text{raw}}(t) = \text{acceleration}(t) * T_s + \text{velocity}(t-1) \tag{1}$$

for each instant $t$, with $T_s = 1/\text{sample frequency}$

$$\text{velocity}_{\text{de-drifted}}(t) = \text{velocity}_{\text{raw}}(t) - \text{sigmoid curve of drift estimation (t)} \tag{2}$$

$$\text{velocity}(t) = \text{velocity}_{\text{de-drifted}}(t) - \text{velocity}_{\text{de-drifted at foot flat}} \tag{3}$$

Note that the initial velocity at the start of the measurement ($t = 0$) was set at zero. Since the measurements started while participants were walking and the leg could be in the swing phase, this could result in an inaccurate velocity estimation until the first foot flat phase was reached.

Numerical integration of the velocity over the duration of the trial (120 s) results in tri-axial position estimation over the duration of the trial:

$$\text{position}(t) = \text{velocity}(t) * T_s + \text{position}(t-1) \tag{4}$$

for each instant $t$, with $T_s = 1/\text{sample frequency}$

Note that the initial position at the start of the measurement ($t = 0$) was set at zero.

Stride time was defined as the time between two consecutive IC events:

$$\text{stride time}_n = \text{time at IC}_n - \text{time at IC}_{n-1} \tag{5}$$

Stride length was defined as the distance traveled by the foot during the stride time (IC till following IC) in the horizontal plane:

$$\text{stride length}_n = \text{sqrt}((\text{positionX}_{\text{ICn}} - \text{position X}_{\text{ICn}-1})^2$$
$$+ (\text{position Y}_{\text{ICn}} - \text{position Y}_{\text{ICn}-1})^2) \tag{6}$$

Stride velocity was calculated as the stride length divided by the stride time:

$$\text{stride velocity}_n = \text{stride length}_n / \text{stride time}_n \tag{7}$$

Gait event detection in the OMCS data was performed according to the validated velocity based method of *Zeni, Richards & Higginson (2008)*. This method defines TC at the instant that the velocity vector in anterior-posterior direction of the toe marker crosses zero in the anterior direction. IC is defined at the instant that the velocity vector in the anterior-posterior direction of the heel marker crosses zero in the posterior direction. For

treadmill walking, the position of the toe and heel markers in the global coordinate system were used whereas the position of toe and heel markers were calculated relative to the pelvis for overground walking. Stride time and stride velocity were calculated according to the same definitions as used for the sensor algorithm (Eqs. (5) and (7)). Stride length for OMCS data during treadmill walking was calculated as the average velocity of the ankle on the contralateral side during flat foot (Eqs. (8) and (9)), multiplied by the stride time and added to the difference in position between IC and the following IC along the $Y$-axis, which is the axis in line with the walking direction (Eq. (10)). In overground walking the stride length was calculated as the difference in position between two consecutive IC events of the heel marker (Eq. (11)).

$$\text{swing time}_n = \text{IC}_n - \text{TC}_{n-1} \tag{8}$$

$$\text{velocity}_n^{\text{treadmill}} = (\text{position } Y_{\text{TC}+0.1*\text{swing time}}^{\text{contralateral foot}} - \text{position } Y_{\text{TC}+0.6*\text{swing time}}^{\text{contralateral foot}})/ \\ (0.5*\text{swing time}) \tag{9}$$

$$\text{stride length}_n \text{OMCS treadmill walking} = (\text{position } Y_{\text{IC}n} - \text{position } Y_{\text{IC}n-1}) + \\ \text{velocity}_n^{\text{treadmill}} * \text{stride time}_n \tag{10}$$

$$\text{stride length}_n \text{OMCS overground walking} = (\text{position } Y_{\text{IC}n} - \text{position } Y_{\text{IC}n-1}) \tag{11}$$

## Post-hoc analysis

Two methods are frequently used in literature to identify gait events: the OMCS-based method used in this study and a method based on force plate data (*Pacini Panebianco et al., 2018*; *Behboodi et al., 2019*; *Mariani et al., 2010*; *El-Gohary et al., 2014*). The benefit of OMCS-based gait event detection is that multiple strides per stretch in the overground lab can be analyzed against only one stride per stretch on force plates. To maximize the number of strides for analysis in the overground lab and be consistent in the methods used, the IMU-based algorithm was validated against OMCS in both settings. Nevertheless, during treadmill walking trials, force data was collected by the embedded force plates of the GRAIL. We checked the magnitude of the difference, including limits of agreement (LoA) at 1.96 standard deviation (SD), in gait event detection between the OMCS-based method and force plate data as ground truth.

## Statistical analysis

Groups were compared on gender distribution by the Chi-square test, and on age, height and weight by the Mann–Whitney-U test. The validity of the gait algorithm was evaluated on a stride-by-stride basis, quantifying the agreement of the instant of IC and TC, stride time, stride length, and stride velocity, with OMCS-derived outcomes as reference (*Zeni, Richards*

*& Higginson, 2008*). For stride time and length variability, we calculated the coefficient of variation (CoV) for each participant, defined as the SD over all strides divided by the mean of all strides within a participant. Differences between sensor and OMCS-derived timing of IC and TC were visualized in histograms. For stride time, stride length and stride velocity, we created Bland-Altman-like plots to reflect the agreement between the IMU-based and OMCS-based analysis. Because the difference between methods for stride length and velocity showed a downward trend with increasing means of the value (non-uniformity), evaluated using linear regression models, we did not calculate the limits of agreement. To evaluate variance within and between subjects, we constructed Bland-Altman-like plots based on the mean over strides within a participant, as well as based on all separate strides (except for CoV measures which can only be calculated per participant).

To evaluate the algorithm's performance in irregular walking, we compared differences between methods for regular with irregular conditions in healthy controls and for regular with irregular conditions in stroke participants using a linear mixed model with the difference between methods for each gait parameters as dependent measure, condition as fixed effect and participant ID as random effect. We also tested differences between methods comparing healthy *versus* stroke participants during regular walking. For this comparison, we constructed linear mixed models with the difference between methods of each gait parameter as dependent variable, group (healthy *vs* stroke) as fixed effect and participant ID as random effect. For CoV measures, we compared regular *vs* irregular walking in healthy participants using a paired $t$-test and compared healthy and stroke participants during regular walking using an unpaired $t$-test.

The significance level was set at alpha 0.05. Differences between overground and treadmill walking in the differences between methods were described by mean differences and SD. All statistical analysis was done in RStudio (R version 2022.02.0; *RStudio Team, 2022*), using the lme4 package (version 1.1-29; *Bates et al., 2015*).

# RESULTS

## Treadmill walking

All participants performed all regular and irregular walking conditions except for one individual with stroke (participant ID: STR_03), whose walking capacity was insufficient to perform the target stepping task. Therefore, only a fixed-speed trial representing regular walking from this participant was included for further analysis. One other stroke participant (participant ID: STR_09) had to perform the regular walking task at a fixed treadmill speed. All other participants performed the regular walking condition in self-paced mode. Stride time varied between 0.71 and 2.58 s, stride length between 0.26 and 1.83 m, and stride velocity between 0.14 and 1.73 m/s across all participants and conditions (Table 2).

## Gait event detection

Detection of IC when collapsing groups and conditions was on average 9-17 ms later based on IMU compared to OMCS (Figs. 2A, 2E). TC was on average 15–24 ms earlier for the IMU-based method (Figs. 2B, 2F). For both gait events, the variance of difference between

**Table 2  Median and IQR of IMU-based and OMCS-based parameters.** Median and IQR were calculated over the mean per trial for each parameter, CoV was calculated as the median and IQR over the CoV per trial for each parameter.

| | Healthy participants ($n = 20$) | | | | Stroke participants ($n = 10$) | | | |
| | Regular walking | | Irregular walking | | Regular walking | | Irregular walking | |
| | IMU | OMCS | IMU | OMCS | IMU | OMCS | IMU | OMCS |
|---|---|---|---|---|---|---|---|---|
| Stride time (s) median (IQR) | 1.05 (0.09) | 1.05 (0.09) | 1.07 (0.13) | 1.07 (0.13) | 1.42 (0.36) | 1.42 (0.36) | 1.36 (0.28) | 1.36 (0.29) |
| CoV stride time (%) median (IQR) | 1.80 (0.66) | 1.80 (0.62) | 6.15 (1.65) | 6.11 (1.45) | 5.62 (3.42) | 4.65 (3.91) | 6.49 (3.42) | 5.79 (3.91) |
| Stride length (m) median (IQR) | 1.32 (0.13) | 1.35 (0.13) | 1.31 (0.10) | 1.34 (0.13) | 0.78 (0.38) | 0.78 (0.38) | 0.76 (0.32) | 0.76 (0.31) |
| CoV stride length (%) median (IQR) | 4.55 (1.98) | 4.74 (2.58) | 6.62 (0.75) | 7.25 (0.77) | 9.20 (3.60) | 8.54 (3.85) | 8.57 (3.60) | 8.51 (3.85) |
| Stride velocity (m/s) median (IQR) | 1.26 (0.12) | 1.29 (0.13) | 1.23 (0.13) | 1.26 (0.16) | 0.62 (0.27) | 0.62 (0.28) | 0.62 (0.39) | 0.61 (0.39) |
| CoV stride velocity (%) median (IQR) | 5.26 (2.62) | 5.48 (3.11) | 5.13 (0.99) | 5.51 (0.98) | 9.39 (2.64) | 8.93 (2.46) | 7.75 (2.64) | 7.61 (2.46) |

methods (SD for each individual) was limited in healthy participants, and more apparent in stroke participants (Figs. 2C, 2D and 2G, 2H).

When comparing regular with irregular walking in healthy participants, the difference between methods for IC was 2.5 (95% CI [2.2–2.8]) ms smaller in the regular condition ($t = 15.76$, $p < 0.001$, Table 3). The difference between methods for TC was 3.4 (95% CI [3.0–3.8]) ms smaller in the irregular compared to regular condition ($t = 18.06$, $p < 0.001$, Table 3). The second operationalization of the effect of irregular walking, comparing stroke with healthy participants, showed that the difference between methods for IC was 7.6 (95% CI [−14.0 to −1.1]) ms smaller for stroke than healthy participants (t = −2.30, $p = 0.029$, Table 3), while TC did not differ significantly between groups (95% CI [−21.2–3.5], t = −1.41, $p = 0.169$, Table 3). The third operationalization of the effect of irregular walking, comparing regular with irregular walking conditions in stroke participants, showed that the difference between methods for IC was 2.6 ms (95% CI [−4.9 to −0.3]) lower for the irregular than regular condition (t = −2.17, $p = 0.030$, Table 3). The difference between methods for TC was 2.2 ms (95% CI [−4.2 to −0.1], t = −2.05 $p = 0.040$, Table 3).

## Spatiotemporal parameters

Figure 3 shows the Bland-Altman-like plots for the spatiotemporal gait parameters averaged per subject (left panels) and on a stride-by stride basis (middle and right panels). The stride time difference between methods did not vary as a function of the value of the method itself in both healthy and stroke participants. Differences between methods for stride time were on average 0 ms, with low between-subject variance for all conditions in healthy participants (SD = 0.01 s) and 0.04 s during regular walking and 0.05 s during irregular walking in stroke participants (Table 4).

In healthy participants, the difference between methods on stride time was not different between regular and irregular walking ($t = 0.153$, $p = 0.878$, Table 3), and not different between healthy participants and stroke participants during regular walking (t = −0.111,

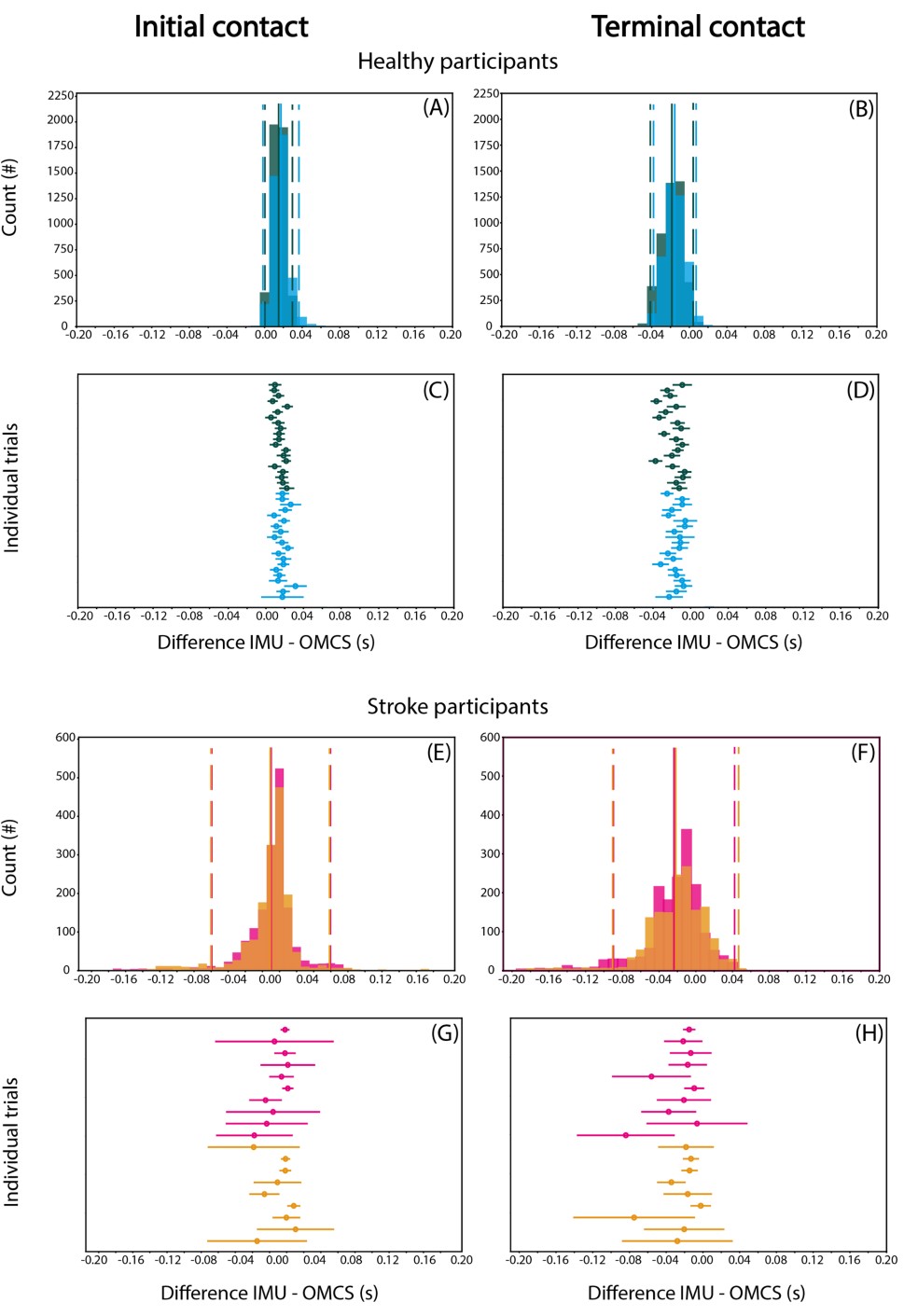

**Figure 2** Differences in IC (A, C, E, G) and TC (B, D, F, H) between IMU-based and OMCS-based algorithms in regular (green) and irregular (blue) walking conditions in healthy participants (top panels, A-D) and in regular (pink) and irregular (orange) walking conditions. Histograms are on a stride-by-stride basis for all participants. Solid vertical lines indicate mean difference and dashed vertical lines indicate the 1.96*SD. The 'Individual trials' plots show the mean difference and SD for each trial.

**Table 3 Statistical output of linear mixed regression models to compare irregular *vs* regular walking in healthy participants and in stroke participants, and comparison stroke *vs* healthy participants in regular walking.** Linear mixed models were used to evaluate the performance of the algorithm in irregular walking compared to regular walking. In the comparison healthy irregular *vs* healthy regular walking, and in the comparison stroke irregular *vs* stroke regular walking, the regular walking trials were used as the reference with the difference between methods for each gait parameter as dependent measure, condition as fixed effect and participant ID as random effect. In the comparison stroke *vs* healthy in regular walking, the walking trials of the healthy participants were used as the reference with the difference between methods of each gait parameter as dependent variable, group (healthy *vs* stroke) as fixed effect and participant ID as random effect.

| | | Intercept | 95% CI | | Coefficient | 95% CI | | $t$-value[*] | $p$-value[*] |
|---|---|---|---|---|---|---|---|---|---|
| | | | Lower | Upper | | Lower | Upper | | |
| Healthy, irregular *vs* regular walking | IC detection | 1.502 | 1.293 | 1.712 | 0.248 | 0.217 | 0.279 | 15.762 | 0.000 |
| | TC detection | −1.879 | −2.223 | −1.534 | 0.339 | 0.302 | 0.376 | 18.057 | 0.000 |
| | Stride time | 0.000 | −0.000 | 0.000 | 0.000 | −0.000 | 0.000 | 0.153 | 0.878 |
| | Stride length | −0.034 | −0.049 | −0.019 | 0.003 | 0.002 | 0.004 | 4.500 | 0.000 |
| | Stride velocity | −0.033 | −0.046 | −0.019 | 0.004 | 0.002 | 0.005 | 5.951 | 0.000 |
| Stroke *vs* healthy, regular walking | IC detection | 1.499 | 1.129 | 1.870 | −0.755 | −1.400 | −0.111 | −2.297 | 0.029 |
| | TC detection | −1.878 | −2.590 | −1.166 | −0.890 | −2.124 | 0.345 | −1.413 | 0.169 |
| | Stride time | 0.000 | −0.001 | 0.001 | −0.000 | −0.001 | 0.001 | −0.111 | 0.912 |
| | Stride length | −0.034 | −0.047 | −0.022 | 0.038 | 0.016 | 0.060 | 3.422 | 0.002 |
| | Stride velocity | −0.032 | −0.044 | −0.021 | 0.034 | 0.014 | 0.054 | 3.313 | 0.003 |
| Stroke, irregular *vs* regular walking | IC detection | 0.876 | −0.080 | 1.833 | −0.256 | −0.487 | −0.025 | −2.173 | 0.030 |
| | TC detection | −2.746 | −4.325 | −1.166 | −0.217 | −0.424 | −0.010 | −2.050 | 0.040 |
| | Stride time | −0.000 | −0.002 | 0.002 | 0.001 | −0.002 | 0.004 | 0.618 | 0.537 |
| | Stride length | 0.003 | −0.004 | 0.011 | 0.001 | −0.002 | 0.005 | 0.652 | 0.515 |
| | Stride velocity | 0.001 | −0.006 | 0.007 | 0.002 | −0.001 | 0.004 | 0.429 | 0.153 |

**Notes.**

IC, Initial contact; TC, Terminal contact.

[*]Relate to the coefficient (not the intercept).

$p = 0.912$, Table 3). Also, no differences between methods were found between the regular and irregular walking condition on stride time in stroke participants ($t = 0.618$, $p = 0.537$, Table 3). Differences between methods for CoV of stride time did not significantly differ between regular and irregular walking in healthy participants ($t = 1.189$, $p = 0.249$, Table 5 and Fig. 4), or between healthy and stroke participants during regular walking (t $= −1.909$, $p = 0.089$, Table 5 and Fig. 4), or between regular and irregular walking conditions in stroke participants (t $= −1.038$, $p = 0.330$, Table 5). However, a larger mean CoV of stride time and stride length in the irregular trials, suggests that the proposed method to induce irregularity seemed to work.

In both conditions, stride length in healthy participants was 0.03 m (SD regular: 0.04 m, SD irregular: 0.05 m) smaller when based on IMUs compared to OMCS (Table 4 and Fig. 3). In stroke participants, the stride length difference between methods was 0.00 m (SD regular: 0.06 m, SD irregular: 0.04 m, Table 4 and Fig. 3). Comparing regular *vs* irregular walking in healthy participants, resulted in larger differences between methods for the regular walking condition (0.003 m, $t = 4.500$, $p < 0.001$, Table 3). The difference between methods for stride length during regular walking was closer to zero for stroke patients compared to healthy participants (0.038 m, $t = 3.422$, $p = 0.002$, Table 3). The difference between methods for CoV of stride length was larger in irregular walking compared to

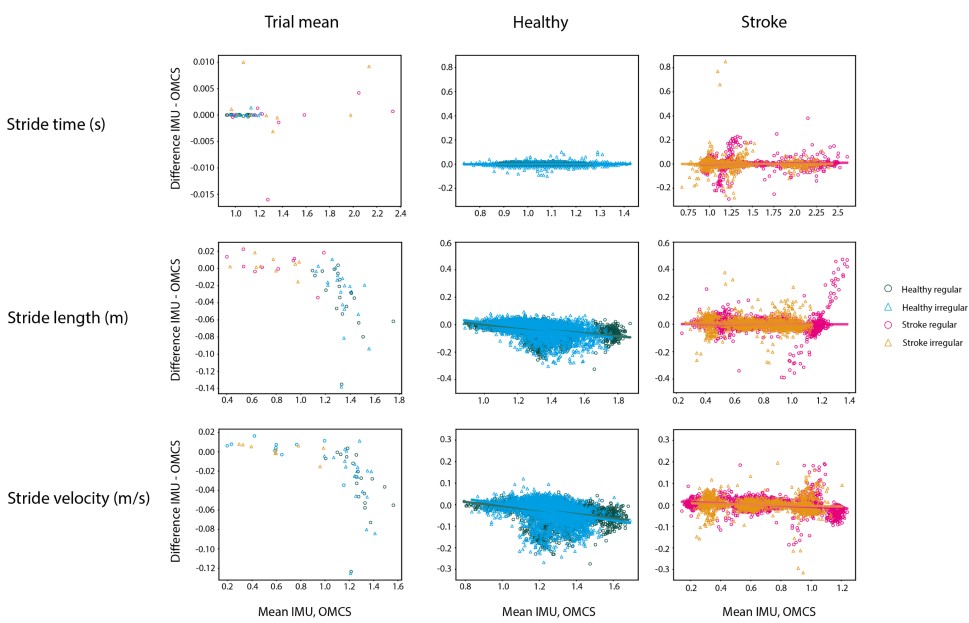

**Figure 3** **Bland-Altman analyses of the mean spatiotemporal parameters per condition in healthy regular (green O), healthy irregular (blue Δ), stroke regular (pink O) and stroke irregular (orange Δ) walking conditions.** Middle panels are the Bland-Altman analyses in the healthy population in regular (green) and irregular (blue) walking on a stride-by-stride basis. Right panels are the Bland-Altman analyses in the stroke population in regular (pink) and irregular (orange) walking on a stride-by-stride basis. Note that the $y$-axis for means per trial is on a different scale as the plots on a stride-by-stride basis.

**Table 4** **Mean and SD of the difference between methods (IMU $vs$ OMCS) during treadmill walking on a stride-by-stride basis.** Differences were calculated as 'IMU-based parameter–OMCS-based parameter' and displayed as mean (SD). IC, initial contact; TC, terminal contact.

| | Healthy participants ($n = 20$) | | Stroke participants ($n = 10$) | |
|---|---|---|---|---|
| | Regular walking | Irregular walking | Regular walking | Irregular walking |
| N strides (total) | 4,577 | 4,200 | 1,671 | 1,586 |
| IC detection (ms) | 15 [7] | 17 [10] | 10 [35] | 9 [35] |
| TC detection (ms) | −19 [12] | −15 [12] | −24 [34] | −22 [35] |
| Stride time (s) | −0.00 [0.01] | 0.00 [0.01] | −0.00 [0.04] | 0.00 [0.05] |
| Stride length (m) | −0.03 [0.04] | −0.03 [0.05] | 0.00 [0.06] | 0.00 [0.04] |
| Stride velocity (m/s) | −0.03 [0.04] | −0.03 [0.04] | −0.00 [0.03] | 0.00 [0.03] |

regular walking (−0.44%, t = −4.198, $p < 0.001$, Table 5). Differences between methods for CoV of stride length in stroke participants (0.7% higher in IMU $vs$ OMCS) was not different from healthy participants during regular walking (0.2% lower in IMU $vs$ OMCS; t = −1.186, $p = 0.266$, Table 5). There were also no differences found between methods for CoV of stride length in irregular walking compared to regular walking in stroke participants (t = −1.165, $p = 0.278$, Table 5).

Stride velocity in healthy participants was 0.03 m/s (SD = 0.04) lower when based on IMUs compared to OMCS (Table 4). In stroke participants, stride velocity difference

**Table 5 Statistical output of t-tests to compare difference between methods for variance (CoV) in stride time and stride length.** Paired samples t-tests were used to evaluate the performance of the algorithm in irregular walking compared to regular walking within healthy participants and within stroke participants. In these comparisons the regular walking trials were used as the reference. Unpaired t-tests (Welch two sample) were used to evaluate the performance of the algorithm in stoke participants compared to healthy participants in regular walking. In this comparison the walking trials of the healthy participants were used as the reference.

| | | Mean difference | 95% CI Lower | 95% CI Upper | *t*-value | *p*-value |
|---|---|---|---|---|---|---|
| Healthy, irregular *vs* regular walking | Stride time | 0.046 | −0.035 | 0.127 | 1.189 | 0.249 |
| | Stride length | −0.439 | −0.657 | −0.220 | −4.198 | 0.000 |
| Stroke *vs* healthy, regular walking | Stride time | 0.994 | −2.172 | 0.184 | −1.909 | 0.089 |
| | Stride length | 1.052 | −3.055 | 0.951 | −1.186 | 0.266 |
| Stroke, irregular *vs* regular walking | Stride time | −0.513 | −1.654 | 0.627 | −1.038 | 0.330 |
| | Stride length | −0.903 | −2.691 | 0.885 | −1.165 | 0.278 |

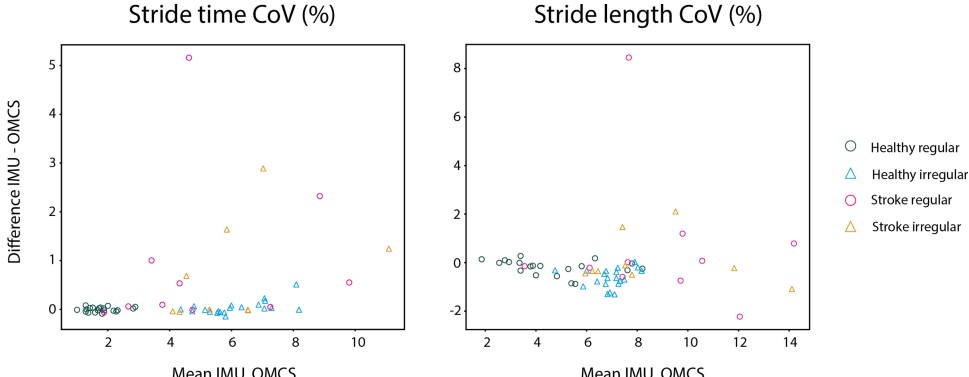

**Figure 4 Bland-Altman analyses of the variability of spatiotemporal parameters per trial in healthy regular (green O), healthy irregular (blue Δ), stroke regular (pink O) and stroke irregular (orange Δ) walking conditions.**

between methods was 0.00 m/s (SD = 0.03, Table 4). Comparing regular *vs* irregular walking in healthy participants, resulted in smaller differences in regular walking (0.004 m/s, $t = 5.951$, $p < 0.001$, Table 3). Differences between methods were larger for healthy participants compared to stroke participants during regular walking (0.034 m/s, $t = 3.313$, $p = 0.003$, Table 3).

Details of the statistical output can be found in Tables 3 and 5. A table including the mean differences and SDs for each subject, for each walking condition can be found in the supplementary materials.

## Overground walking

Spatiotemporal parameters per subject ranged between 0.94 and 1.28 (median = 1.01) s for stride time, 1.18 and 1.64 (median = 1.40) m for stride length and 1.05 and 1.70 (median = 1.36) m/s for stride velocity. Table 6 shows differences in gait event detection

**Table 6  Mean differences and SD between IMU-based and OMCS-based parameters during overground walking.** Differences were calculated as IMU-based parameter–OMCS-based parameter' and displayed as mean [SD].

| Healthy participants ($n = 20$) | Overground walking |
| --- | --- |
| N strides (total) | 1,426 |
| IC detection (ms) | −6 [19] |
| TC detection (ms) | −40 [17] |
| Stride time (s) | 0.00 [0.03] |
| Stride length (m) | −0.08 [0.05] |
| Stride velocity (m/s) | −0.08 [0.07] |

and spatiotemporal parameters between IMU-based and OMCS-based analysis during overground walking.

### Post-hoc analysis

OMCS detected IC 0.03 s [LoA: −0.01; 0.07] and TC 0.01 s [LoA: −0.03; 0.05] after the force plates. See Supplementary Materials for full details of this analysis and histograms (Fig. S1) of the mean differences.

## DISCUSSION

The aim of this study was to evaluate the developed algorithm for gait assessment using IMUs on both feet and the trunk, for regular as well as irregular walking patterns. We found high accuracy of gait event detection, stride time, and stride time variability during regular and irregular walking in healthy participants compared to OMCS. In healthy participants, mean stride length and stride velocity were slightly underestimated with three cm and three cm/s, respectively. However, the accuracy was much worse in several healthy participants, with errors increasing up to 13 cm and 13 cm/s, respectively. The algorithm's accuracy did not substantially worsen for irregular walking compared to regular walking in healthy participants. Likewise, the irregular walking pattern that was observed in stroke participants resulted in similarly high accuracy of the algorithm.

The accuracy of our IMU-based algorithm on stride time in healthy participants was $0 \pm 10$ ms, which was comparable to previous research evaluating accuracy during regular walking with errors of $9 \pm 22$ ms (*Morris et al., 2019*). Regarding spatial parameters, previous validation studies with sensors on the feet have reported an average underestimation of stride length between 2 and 12 cm in healthy participants (*Hundza et al., 2014*; *Morris et al., 2019*; *Zhu, Anderson & Wang, 2012*). In the study by Morris et al., an average underestimation of 10 cm with their IMU-based algorithm was reported, which increased to 18 cm with increasing stride length. This trend of increasing underestimation with increasing stride length, and thus increasing gait speed, was also seen in the current study. At gait speeds above 1.2–1.3 m/s, the underestimation of stride length increased to maximally 13 cm in one subject. Although this error is still smaller than as reported in *Morris et al. (2019)*, increasing errors with increasing gait speeds is a significant concern when applying foot-mounted IMU algorithms for the assessment of healthy gait. Caution

should also be warranted when comparing groups with different gait speeds. When verifying these results for the overground trials, a slightly larger but still acceptable error of 8 cm compared to 10 cm reported in previous literature was found (*Morris et al., 2019*).

In artificially induced irregular walking in healthy participants by irregularly spaced stepping targets, the algorithm's accuracy was similar to regular walking. The higher mean CoVs of stride time and stride length in the irregular walking condition compared to the regular walking condition indicated that the irregular walking manipulation was successful. In contrast, the irregular walking condition in stroke participants slightly increased the CoV of stride time, but did not impact the CoV of stride length. No differences in the accuracy of stride time estimation were found between the irregular and regular walking conditions (0 ms) in both groups. Although significant, the differences in accuracy between irregular and regular walking were only 0.3 cm for stride length and 0.3 cm/s for stride velocity in healthy participants. Therefore, we concluded that temporal and spatial parameters were assessed with the same accuracy in irregular walking compared to regular walking.

To be able to use the gait algorithm in clinical populations, evaluation of the algorithm's performance in irregular walking patterns due to pathology was an important aim of this study. In the stroke population, the mean error of the estimated stride length compared to OMCS was 0 cm and 0 cm/s for stride velocity. In previous research, the calculated error for stride length in people with irregular gait due to Parkinson's disease was also lower than in healthy participants (*Morris et al., 2019*) (Parkinson's disease group 8.5 cm *vs.* healthy peers 10 cm).

The higher accuracy in people with stroke compared to healthy participants was not in line with our hypothesis. One factor underlying a higher accuracy of spatial parameters in stroke participants might be the slower walking speed in this group compared to the healthy group. As stride length was calculated by double integration of the acceleration data, relatively small errors in event detection or timing of zero-velocity updates can cause inflated errors in spatial parameters. Consequently, lower walking speeds resulting in lower acceleration peaks are less affected by errors in gait event detection compared to faster walking speeds with higher acceleration. However, IC and TC event detection were highly accurate in healthy participants with low between-strides variance, at least partly contradicting this explanation. Therefore, a small error in the timing of zero-velocity updates seems most likely, as we did not validate the detection of the foot flat phase (TC to mid-swing of the contralateral leg). Additionally, between foot flat phases, a drift compensation based on a sigmoid curve is performed. This drift compensation might overestimate the actually measured drift, this results in subtracting too much of the acceleration leading to an underestimation of stride length and velocity. Exploring other drift compensation techniques might further improve the accuracy of the algorithm.

In addition to mean values of gait parameters, variability between the strides of an individual (CoV) is of clinical interest (*Hausdorff, 2009*; *Del Din et al., 2019*). Variability of stride time could be accurately assessed with IMUs in both healthy participants and stroke patients. In contrast, lower accuracy was found in spatially dependent CoV parameters with the IMUs.

This study has some limitations meriting attention. First, all participants walked at their preferred gait speed, resulting in a different range of gait speeds between both groups. Therefore, we cannot distinguish the effect of walking slower from the effect of walking more irregularly. This could be evaluated by having healthy participants walk at lower than comfortable speeds and people with stroke walk at higher gait speeds. A downside of this approach is that it might lead to unnatural gait patterns, reducing the ecological validity of the results. Additionally, we observed that even after a familiarization period of the self-paced mode, some participants had difficulty maintaining a constant comfortable walking speed during the regular walking condition. This most likely resulted in a higher CoV in stride length, time, and velocity in the regular walking condition than reported in the literature (*Kroneberg et al., 2019*). Secondly, we only focused on a limited number of spatiotemporal parameters among many of the potential gait characteristics reported in the literature (*Constantinou et al., 2014*; *Mohan et al., 2021*). The selected spatiotemporal gait parameters are the most crucial in the algorithm to assess spatiotemporal gait parameters. Additional parameters such as step time and double support time are typically derived from the identified gait events and parameters included in this study. Nonetheless, it could be valuable to analyze the accuracy and errors of other spatiotemporal gait parameters when these are used for research or clinical purposes. Thirdly, we designed our protocol to have equal walking duration for all participants and in each condition. Because participants walked with different walking speeds, stride length and time, different number of strides between subjects were recorded. It might be beneficial for future studies to standardize the number of measured strides. Lastly, verification that the IMU-based algorithm could also be used in overground walking was only performed in healthy participants. This was done to decrease the burden on stroke patients, as we had no reason to suspect different results for this analysis in stroke participants, but we cannot provide proof for this assumption.

## CONCLUSIONS

Overall, the accuracy of the proposed IMU-based algorithm was high for temporal gait parameters in regular and irregular walking patterns in healthy and people with stroke, while there was room for improvement for spatially dependent parameters. Although *general* accuracy in irregular gait was as good as in regular walking, stride length and velocity errors in individual cases were substantial and beyond clinically relevant differences. Therefore, the IMU-based algorithm performs satisfactory for walking speeds up until 1.2 m/s. Caution should be applied when considering individual outcomes, groups walking at high gait speed, and when comparing groups with different walking speeds. Further development of algorithms is needed for these purposes.

### Funding

The collaboration project is co-funded by the PPP Allowance made available by Health Holland, Top Sector Life Sciences & Health, to Stichting ReumaNederland to

stimulate public-private partnerships, and by Smith and Nephew, Hoofddorp. The APC was funded by ZonMw (Topspecialistische Zorg & Onderzoek 10070022010004). There was no additional external funding received for this study. The funders had no role in study design, data collection and analysis, decision to publish, or preparation of the manuscript.

### Grant Disclosures

The following grant information was disclosed by the authors:
PPP Allowance made available by Health Holland, Top Sector Life Sciences & Health.
ZonMw (Topspecialistische Zorg & Onderzoek 10070022010004).

### Competing Interests

The authors declare there are no competing interests.

### Author Contributions

- Carmen Ensink conceived and designed the experiments, performed the experiments, analyzed the data, prepared figures and/or tables, authored or reviewed drafts of the article, and approved the final draft.
- Katrijn Smulders conceived and designed the experiments, analyzed the data, prepared figures and/or tables, authored or reviewed drafts of the article, and approved the final draft.
- Jolien Warnar conceived and designed the experiments, performed the experiments, authored or reviewed drafts of the article, and approved the final draft.
- Noel Keijsers conceived and designed the experiments, analyzed the data, authored or reviewed drafts of the article, and approved the final draft.

### Human Ethics

The following information was supplied relating to ethical approvals (i.e., approving body and any reference numbers):

The METC Oost-Nederland granted this study and exemption of the Dutch medical scientific research act (WMO) (identification number: 2021-8191).

### Data Availability

The SintMaartenskliniek/IMU_GaitAnalysis is available at Github and Zenodo:

- https://github.com/SintMaartenskliniek/IMU_GaitAnalysis, Release 'Validation study, tag: v1.1.0'

- Carmen Ensink, Katrijn Smulders, Jolien Warnar, & Noel Keijsers. (2023). Sint Maartenskliniek - IMU Gait Analysis (Validation study, release tag v1.1.0) (v1.1.0) [Data set]. Zenodo. https://doi.org/10.5281/zenodo.8198714.

### Supplemental Information

Supplemental information for this article can be found online at http://dx.doi.org/10.7717/peerj.16641#supplemental-information.

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
