# Peer review of "Validation of an algorithm to assess regular and irregular gait using inertial sensors in healthy and stroke individuals"

_PeerJ, doi:10.7717/peerj.16641_

## Round 0.1 · original submission · Major Revisions

Your above paper has now been reviewed by expert referees, whose comments are enclosed for your perusal. On the basis of these comments, we would like to invite you to revise your paper, taking into account the issues raised by the reviewers. Please note that acceptance is not guaranteed at this stage and any revision is likely to be sent back to the referees for further review. You should, therefore, include a point-by-point response to the reviewers' comments, highlighting each change made in your manuscript and/or providing a suitable rebuttal. Please remember to update the abstract (if appropriate) to reflect any changes made to the manuscript following review.

Reviewer 1 ·

Basic reporting

This study involved validating an algorithm for detecting foot contacts and involved comparison of regular and irregular walking in healthy subjects as well as a comparison between regular walking in healthy individuals vs. stroke survivors. Below is my critical review of the article in the form of comments to the authors, which may be beneficial in further improving the quality of the manuscript prior to acceptance.

1.1. It may be beneficial to introduce motion capture systems, their role in evaluation of human health/performance, types of these systems and currently available IMU-based systems to the reader at the start of introduction, prior to describing the drawbacks of current algorithms.

1.2. Line 14: Please include citations of all the relevant studies that have developed algorithms for gait analysis using IMU-based sensors.

1.3. I commend the authors for making the data and code publicly available for future studies.

1.4. Please consider adding further depth to the literature review in the introduction and discussion section by including relevant studies that have implemented wearable sensors for evaluating movement alterations during walking in healthy and in stroke patients. It may also be helpful to discuss other methods and tools that can be/are being used for evaluation besides IMU-based systems (for example, wearable insole and pressure mat systems). Lastly, the benefits of using an IMU-based system (for instance their portability and ease of evaluation) vs. other methods including optical motion capture system may also be included.

Experimental design

2.1 Line 47: Please indicate whether any of the participant demographics shown in Table 1, between the groups (regular vs. irregular) and (healthy vs. stroke) were statistically significant. If any of them are statistically significant, please include their p-values.

2.2 The experimental and analysis methods have been thoroughly described and illustrated with the help of figures as well as equations.

Validity of the findings

3.1 Figure 3 and Figure 4: It may be better to have different color coding for each of the four categories since identifying the categories based differences in the shapes (circle vs. triangle) for healthy vs. stroke groups is currently difficult.

3.2 Did the authors exclude a portion of initial and final stages of treadmill and overground walking for removing the acceleration and deceleration phases of walking to ensure less variability in the speed? If not, then authors may include this aspect of reaching and staying at a constant self-paced speed in their discussion.

Additional comments

4.1 The authors may consider updating the title of the paper to also include the comparison between healthy and stroke survivors.

4.2 Line 354: Do you mean ‘tested’ instead of ‘teased’?

Reviewer 2 ·

Basic reporting

The English language used in the manuscript is clear
The introduction and background provide context, citing the literature correctly.
All Raw data are supplied with sufficient comments.
Figures can be improved.

See attached file for details.

Experimental design

Methods can be improved.

See attached file for details.

Validity of the findings

The algorithm validation has some weaknesses.
The authors concluded that the IMU-based algorithm performs satisfactorily for walking speeds below 1.25 m/s. I think that this is a great limitation of the algorithm in the context of regular walking. The typical walking speed of healthy adults is about 1.4 m/s, therefore the proposed algorithm cannot be used in the assessment of the “regular” gait in healthy participants.


See attached file for details

Annotated reviews are not available for download in order to protect the identity of reviewers who chose to remain anonymous.

---

## Round 0.2 · Major Revisions

We would like to invite you to revise your paper, taking into account the issues raised by the reviewer.

Reviewer 2 ·

Basic reporting

See attached file

Experimental design

See attached file

Validity of the findings

See attached file

Additional comments

See attached file

Annotated reviews are not available for download in order to protect the identity of reviewers who chose to remain anonymous.

---

## Round 0.3 · Minor Revisions

Please address carefully the changes suggested by the reviewer. Both formulas (1) and (4) are wrong and meaningless. Please note that you can use the integral operator only on continuous functions not on discrete functions (see fundamental theorems of calculus). When you use a definite integral of a function over an interval of time, the lower and the upper boundaries of the integral have to be time values, not sample.

Reviewer 2 ·

Basic reporting

The authors have addressed my concerns. Regarding equations 1 and 4, there is one last point to note. In the integral, acceleration(t) should be used instead of acceleration(i), and velocity(t) should be used instead of velocity(i).

Experimental design

The authors have addressed my concerns.

Validity of the findings

The authors have addressed my concerns.

---

## Round 0.4 · accepted · Accept

I confirm that the article is now Acceptable